# Telomeres and Telomerase in the Control of Stem Cells

**DOI:** 10.3390/biomedicines10102335

**Published:** 2022-09-20

**Authors:** Alexey Yu. Lupatov, Konstantin N. Yarygin

**Affiliations:** Cell Biology Laboratory, Institute of Biomedical Chemistry, Pogodinskaya 10, 119121 Moscow, Russia

**Keywords:** stem cells, cell senescence, replicative lifespan, aging, telomeres, telomerase

## Abstract

Stem cells serve as a source of cellular material in embryogenesis and postnatal growth and regeneration. This requires significant proliferative potential ensured by sufficient telomere length. Telomere attrition in the stem cells and their niche cells can result in the exhaustion of the regenerative potential of high-turnover organs, causing or contributing to the onset of age-related diseases. In this review, stem cells are examined in the context of the current telomere-centric theory of cell aging, which assumes that telomere shortening depends not just on the number of cell doublings (mitotic clock) but also on the influence of various internal and external factors. The influence of the telomerase and telomere length on the functional activity of different stem cell types, as well as on their aging and prospects of use in cell therapy applications, is discussed.

## 1. Introduction

Stem cells are one of the favorite objects of biomedical research. The number of scholarly papers devoted to stem cell studies has been steadily expanding for over forty years (https://pubmed.ncbi.nlm.nih.gov (accessed on 10 March 2022)). Stem cell research is shedding light on one of the main mysteries of metazoan biology—how one cell, a fertilized egg, gives rise to all the diversity of cells in an adult multicellular organism. It has already provided the basis for a number of advanced technologies, such as the breeding of transgenic animals, including those tolerant to diseases [1] or producing biologically active substances [2], somatic cell nuclear transfer (SCNT)-based animal cloning [3], and in vitro production of bioartificial tissues and organs [4]. Stem cell transplantation (stem cell therapy) and stem cell-based tissue engineering are the keystones of regenerative medicine, an innovative trend in medicine with the potential to become a new healthcare industry [5,6,7,8,9]. New stem cell therapy applications, such as hematopoietic and mesenchymal stem cell transplantation in patients with hematological malignancies or autoimmune diseases, are emerging [10,11].

Stem cells constitute a stock of cell material used for growth, development, and regeneration of the organism. The ability to supply cells required to maintain growth and regeneration is largely determined by their proliferative potency, i.e., the number of mitoses they can go through. Each mitosis reduces the proliferative potency of stem cells leading to their replicative senescence and contributes to the aging of the whole organism. However, organismal aging and stem cell senescence are interrelated in a sophisticated way [12,13,14]. A better understanding of the connection between replicative senescence and organismal aging requires insight into stem cell biology, since stem cells are crucially important for the lifelong maintenance of the integrity of the organism.

Clinical applications of stem cells also require an evaluation of their replicative age. The limited ex vivo lifespan can make the preparation of stem cell-based products for biomedicine difficult or impossible. Besides, it is important to know how the cells will behave after they are transplanted to the patient in order to anticipate their effectiveness and the risk of malignant transformation. Here, we evaluate the available data concerning the role of the replicative senescence of stem cells in the disruption of the organism’s integrity and consider the impact of the replicative senescence of different types of stem cells with the emphasis on the role of telomeres and telomerase.

## 2. What Are Stem Cells?

Currently, there is a trend towards an extended interpretation of the term “stem cell”. Quite often, a cell is termed “stem cell” merely because it has not yet undergone terminal differentiation and can still proliferate. This situation highlights the importance of the accurate identification of the true stem cell hallmarks.

The term “stem cells” is commonly applied to a self-renewing population of undifferentiated or minimally differentiated cells capable of producing more differentiated progeny. Fetal hematopoietic stem cells were the first stem cells discovered and characterized [15]. The research community’s and the public’s interest in stem cell studies surged manyfold after the discovery of cells capable of in vitro differentiation into any type of cell in the organism. These cells, named embryonic stem cells (ESCs), were isolated from early embryos at the end of the 20th century [16,17]. ESC transplantation into the blastocyst, followed by the blastocyst implantation into the uterus, provided a way to raise animal chimeras. The ability of ESCs, including those genetically modified, to produce germline cells provided a method to select fully transgenic animals through inbreeding [18]. In parallel with ESC studies, various more specialized stem cells, serving as precursors of different cell types, were discovered and characterized in fetuses and adult organisms. Adult stem cells have noticeable differences from embryonic stem cells in their differentiation and proliferative capacity, including susceptibility to cellular senescence [19]. At present, stem cell concept and further stem cell research is crucial for both basic research and biotechnological applications.

Stem cells can be classified by the stage of ontogeny they are in and by their differentiation potency. **Totipotent cells** can generate both embryonic and extraembryonic tissues. In humans, totipotency is restricted to fertilized oocytes and their progeny after the first two cell divisions. **Pluripotent cells** have the capacity to differentiate into all embryonic but not extraembryonic tissues. ESCs isolated from the blastocyst inner cell mass (ICM) [20] and the induced pluripotent stem cells (iPSCs) [21]—generated via the reprogramming of somatic cells by transfecting them with the so-called pluripotency genes—are pluripotent. In ontogenesis, the ICM pluripotent cells can undergo triploblast differentiation, i.e., can produce all three germ layers—the ectoderm, the endoderm, and the mesoderm, and, consequently, all cell types derived from the three germ layers. Stable ESC populations likely exist only in vitro, since in vivo the commitment of the ICM cells to one of the germ layers takes just 1–2 days [20]. Some researchers have reported the isolation of pluripotent cells from adult tissues [22], including bone marrow [23]. However, the ability of these cells to undergo triploblast differentiation in vivo has never been firmly proved, and the reported data may be explained by stem cell plasticity in culture [24].

During histo- and morphogenesis, cells differentiate and assemble into specific histological and anatomical structures. However, a relatively small fraction of cells, postnatal or adult stem cells, avoid differentiation into cells carrying out specific functions. Therefore, adult tissues contain a plethora of stem cells with varying differentiation potential. Most of the time, stem cells are confined to special tissue-specific compartments, stem cell niches, discovered back in the 1970s [25], where they reside in quiescent state. Under the influence of signals from surrounding cells in the niche, extracellular matrix, blood, tissue fluid, and interstitial nerve elements, stem cells can undergo activation, enter mitosis, and differentiate [26,27]. Adult stem cells have different differentiation potencies and can be **unipotent**, capable of differentiating into cells of a single type (e.g., stem cells of the stratified squamous epithelium of the skin); **oligopotent**, able to form two or more lineages within a specific tissue (e.g., fetal neural stem cells (NSCs) differentiating into neurons or glia); and **multipotent**, capable of differentiating into different cell types derived from a single germ layer (e.g., bone marrow HSCs giving rise to all blood cells).

Besides generating progeny capable of differentiating into functional cell types, true stem cells should show the ability to endure as a separate cell population. Since stem cell populations persist throughout the life of the organism, they should have a prolonged lifespan. This is achieved via the unique ability of stem cells to undergo asymmetrical division yielding two different daughter cells: one entering differentiation and eventually moving towards its new location, and the other remaining identical to the maternal stem cell with regard to phenotype and location [28]. Depending upon the incoming stimuli, stem cells can also undergo normal symmetrical mitosis yielding two stem cells or two differentiating cells [29]. The lifelong self-reproduction of stem cell populations distinguishes stem cells from progenitor cells, their immediate descendants proceeding further with differentiation.

The inability to exist outside a community of similar cells within the microenvironment of a special tissue niche is another stem cell hallmark. Different stem cell types dwell within corresponding niches, such as the HSC niche in the bone marrow [30] or the limbal epithelial stem cell niche in the cornea [31]. Stem cells get various signals from their microenvironment through contact with stromal cells and the extracellular matrix and via receptors specific to soluble signal molecules. They also react to physical stimuli, such as hypoxia or sheer stress. The environmental signaling determines stem cells’ fate: to stay quiescent or to replicate, to undergo symmetrical or asymmetrical mitosis, and what differentiation route to follow [32]. Because of the dependence of stem cells upon the niche, their maintenance in an undifferentiated state in culture requires modeling of the niche signaling, usually by growing stem cells in the presence of feeder cells and/or by conditioning the culture medium with special additives.

Since stem cells form a repository of cells critical for the organism’s survival, they need the utmost protection against damaging agents. Accordingly, they usually display high expression of protective proteins, such as aldehyde dehydrogenases (ALDH) catalyzing the detoxification of aldehydes [33] or ABC-transporters and other cellular membrane pumps ensuring the removal of xenobiotics from the cytoplasm. The activity of cellular membrane pumps can be measured by flow cytometry after loading cells with a vital dye revealing the «side population» (SP) consisting of cells actively excluding the dye [34]. Mouse bone marrow HSC populations identified by surface marker expression and SP analysis were found to match [35]. This was demonstrated also for stem cells isolated from some other murine tissues [36]. However, cellular membrane transporters are not universal stem cell markers. Thus, ESCs isolated from human blastocysts, unlike their murine counterparts, do not form an SP [37]. Moreover, we found that a substantial part of cancer stem cells isolated from tumors of colorectal cancer patients by the expression of the CD133 cancer stem cell marker resisted vital dye staining, and this was not because of the efflux of the dye from the cells [38].

So far, no universal stem cell molecular markers have been revealed [20]. Oct-4, NANOG, SOX2, and other transcription factors ensuring the maintenance of the pluripotent state are hardly necessary to maintain functionality of adult stem cells with limited differentiation capacity. Adult stem cells are usually identified by sets of surface markers characterizing different stem cell types. Since, by definition, any type of stem cell needs substantial proliferation capacity, telomerase emerges as the most relevant and universal stem cell marker.

## 3. Proliferative Senescence and Telomere Shortening: From Mitotic Clock to Molecular Biosensor

Back in 1912, based on his innovations in tissue culture techniques, Alexis Carrel claimed that live cells could be maintained in vitro for an indefinitely long period, meaning that cells are actually immortal [39]. This belief was generally shared by other researchers throughout the first half of the 20th century and was based not only on Carrel’s experiments with long-term chicken heart culture but also on contemporary tissue culture practice, mostly limited to the maintenance of tumor cells. The first paper describing opposing experimental results and suggesting that cells have a limited proliferative capacity appeared at the end of the 1950s [40]. Shortly after that, Leonard Hayflick and Paul Moorhead cultivated 35 strains of cells isolated from different parts of a human fetus and demonstrated that they could undergo no more than 55 population doublings (PD) before proliferation was halted [41]. Unlike others, Hayflick supposed that these results could reflect a fundamental biological phenomenon, not just a methodological problem, and suggested that they represent the process of cell aging. However, at that time the mechanism underlying this phenomenon remained unresolved.

Among all biomolecules only DNA is sufficiently sustainable to keep record of all past cell divisions over a long time. Alexey Olovnikov speculated that replicative aging might be caused by incomplete end replication of linear DNA caused by the lack of the last RNA primer for the lagging strand replication or the presence of a “catalytically inactive zone” in DNA polymerase enzyme [42]. Indeed, DNA polymerases catalyze elongation of the daughter DNA strand, but not its de novo synthesis. This creates a problem in the DNA replication at the 3′ end for all living organisms from viruses to humans (Figure 1A). Arrangement of the genome in the form of a circular DNA molecule looks like the best solution adopted for short genomes by the prokaryotes and many viruses (Figure 1B). In some DNA viruses with linear genome organization, the problem was solved in an ingenious way (Figure 1C–E). Eukaryotes were less creative and resorted to placing multiple tandem nucleotide repeats (telomeres) shortening at each DNA replication at the ends of the chromosomes [43]. The incomplete end-replication hypothesis was seriously challenged by the shortness of eukaryotic RNA primers comprising just 7–12 nucleotides, which is clearly insufficient to deplete the telomeres after the Hayflick limit of cell divisions [44]. However, it was shown that primases place RNA primers not at the very 3′ end of the daughter DNA strand but 70–100 nucleotides upstream [45].

In vertebrates, telomeres consist of arrays of TTAGGG repeats with a single-stranded 3′ overhang and form a complex spatial structure named T-loop [46]. Interestingly, telomeres of the immortal worms *Schmidtea mediterranea* have identical nucleotide sequences [47]. Telomeres are associated with shelterin, a protein complex that stabilizes chromosome ends and prevents activation of the DNA damage response (DDR) system, reacting to DNA breaks [48]. Telomere DNA is transcribed. RNA polymerase II generates a long noncoding TERRA transcript, which, probably together with telomere-associated proteins, fulfills important functions, including the maintenance of telomere length and regulation of telomere-bound protein composition during cell cycle progression [49]. Sustentation of telomere length, especially important for germline cells, is carried out by telomerase [50,51], a reverse transcriptase comprising a catalytic subunit (telomerase-reverse transcriptase, TERT), an RNA template complementary to TTAGGG (TERC), and several accessory proteins (Figure 2).

Studies of proliferative senescence resulting from telomere shortening could explain not only cell aging but also aging of the whole organism. Incomplete DNA replication and the ensuing telomere shortening cause a loss in the proliferative potential of stem cells. They fail to maintain physiological regeneration of tissues and organs manifested as age-related phenotypic changes [52]. Unlike others, germinative stem cells maintain optimum telomere length and do not undergo aging across generations. This aging theory gained support in the experiments with knockout mice. TERC knockout animals have a shorter lifespan than a wild-type lifespan, decreasing in each successive generation. In contrast, mice overexpressing telomerase live longer, if they do not develop tumors, the incidence of which is substantially higher in these mice [53].

Despite its apparent consistency, the concept described above was promptly challenged. Jerry Shay and Woodring Wright [54] demonstrated that the cessation of proliferation beyond the Hayflick limit might not be a result of critical telomere shortening and in the case of human fibroblasts was controlled by p53 and RB tumor suppressors. Inactivation of these proteins results in the resumption of proliferation lasting for other 20–25 PD. At this stage, telomere shortening becomes critical, and most cells in the culture develop genomic instability and die. On the other hand, enhanced mutagenesis associated with genomic instability leads to the emergence of immortal cells characterized by permanent telomerase activation. Within the framework of this concept, it was suggested that telomere shortening beyond a certain level promotes chromatin remodeling [54] or the loss of protection provided by the shelterin complex, resulting in the activation of the DDR system, including the p53/RB-dependent mechanism of proliferation arrest [55]. There were also other questions about the mitotic clock that needed to be answered.

Despite the short mouse lifespan, their telomere length reaches 10 kb, which is several times more than in their human counterparts [56]. Short telomeres are found not only in humans but also in many big and long-living animals, which is probably associated with an evolutionary acquired mechanism of antitumor defense [57]. In spite of the similarity of molecular machinery providing DNA replication in mice and humans, mouse telomeres shorten almost 100 times faster [58]. Other findings, such as the age-dependent telomere shortening in quiescent tissues [59], also cast doubt on whether the incomplete 3′ end replication of DNA is the only mechanism of telomere shortening.

Now, it is commonly accepted that, besides incomplete replication, a number of other factors, including oxidative stress, exonucleolytic processing, inflammation, environmental factors, and therapeutic interventions, can affect telomere shortening [60,61]. There is evidence indicating that oxidative stress contributes the most to this process [62]. Clinical and animal model studies have revealed a correlation between the manifestation of the oxidative stress markers and telomere length or telomere shortening dynamics [63,64,65]. Mitochondria are the main source of reactive oxygen species (ROS) produced as byproducts of aerobic respiration through the addition of extra electrons to oxygen molecules [66]. Accretion of damaged mitochondria due to defective autophagy can be an important mechanism of ROS accumulation [67]. Oxidative stress occurs when, for different reasons, such as aging, disease, environmental pollution, deficient nutrition, etc., the antioxidant system fails to provide complete ROS inactivation. Telomeres are replete with guanine residues, which are attacked by ROS, yielding 8-hydroxy-2-deoxyguanosine (8-oxodG) and leading to the formation of single-strand breaks interfering with replication [68]. Additionally, 8-oxodG lesions prevent full assembly of the functional shelterin complex, inhibiting the incorporation of the telomeric-repeat-binding factor 2 (TRF2) and the protection of the telomeres 1 (POT1) protein, responsible for the inhibition of two DDR-initiating kinases, the ataxia telangiectasia mutated (ATM) kinase and the Rad3-related (ATR) kinase, respectively [69]. The ATM/ATR kinases identify telomere DNA as damaged and eventually trigger p53, which induces a cell cycle checkpoint to allow time for DNA repair [70]. Due to telomere structural uniqueness, the reparation of DNA lesions in the telomeres can take 20 times more time than in the rest of the genome [62,71]. In this case, the strategy “it is easier to remove than to cure” may be at work, resulting in telomere shortening and erosion, replicative senescence, or apoptosis.

There is evidence that the released telomeric DNA ends up in the cytoplasm, where it interacts with toll-like receptors (TLRs), cyclic GMP-AMP synthase (cGAS), and the stimulator of interferon genes (STING) contributing to the formation of the senescence-associated secretory phenotype (SASP) [72]. SASP is typical for old cells and is characterized by the secretion of large amounts of pro-inflammatory cytokines. SASP is the reason for systemic chronic inflammation in elderly people, called inflammaging, which is the main risk factor for the most common age-related diseases and dysfunctions [73]. The release of the extrachromosomal DNA into the extracellular space can also initiate inflammation mediated by its uptake by the dendritic cells [74]. In addition, telomere erosion directly stimulates inflammation through the activation of the ATM kinase-dependent phosphorylation of the YAP1 protooncogene. YAP1 acts as a transcriptional co-activator of the IL-18 precursor and a number of other genes involved in the inflammasome formation [75]. In its turn, the cytokine-induced inflammation causes oxidative stress, which stimulates telomere attrition and closing the vicious circle.

At the organismal level, chemical pollutants, radiation, infections [76], smoking [77], and even psychological stress [12,78] or apnea [79] can enhance telomere shortening (Figure 3). Retrospective studies have demonstrated that survivors from Leningrad, who experienced famine prenatally or in early childhood during the 872-day-long siege of the city during World War II, had dramatically shortened telomeres after 60+ years [80]. In this and similar cases, negative environmental factors are likely to affect the initial length of telomeres by the attenuation of the telomerase activity during pregnancy and infancy. In contrast, a healthy lifestyle, including moderate physical activity [81] or an antioxidant-rich diet, can slow down telomere shortening [82]. Given this, the substantial increase in telomere length in the astronaut Scott Kelly, who spent almost a year on the international space station under conditions of increased radiation and psychological stress, looks controversial [83]. However, it was a short-term effect and the average telomere length returned to its original value within 6 months after Kelly returned to Earth. Probably, the telomere elongation was due to the telomerase activation in renewing blood cells, in which the telomere length was measured. Importantly, under space conditions, telomere elongation was combined with genomic instability and immunological disorders capable of modulating telomerase expression in actively proliferating blood cells.

In addition to incomplete replication of the 3′ end of DNA, various external and internal factors can cause attrition of telomeres, mainly stimulating the excessive production of mitochondrial ROS. Chronic infections and autoimmunity, ATM kinase-dependent inflammasomes, and altered proteins and other biomolecules, including fragments released during chromosome attrition, can initiate chronic inflammation associated with aging (inflammaging). It can not only increase the ROS concentration but also stimulate proliferation within the stem cell niche, causing cell senescence and stem cell depletion.

Under oxidative stress conditions, since 8-oxodG prevents telomere restoration when incorporated by telomerase, telomere attrition can proceed even if telomerase is expressed [84]. Moreover, TERT can move from the nucleus to the mitochondria, where it binds to mitochondrial DNA and protects it from damage caused by ROS [85].

Compared to incomplete end replication, oxidative stress affects telomere length in a more random way. Accordingly, telomere length within a single cell may vary substantially. While the average telomere length seems to be sufficient for further proliferation, it can be prevented by the crucially short telomeres of some chromosomes. Dysfunctional telomeres can be found even in early passages, resulting in irreversible cell cycle arrest in some cells [86]. In case of prolonged cultivation, the number of such cells increases sharply. Due to the telomere position effect, telomere shortening can affect the expression of genes associated with aging, including the TERT localized in the subtelomeric region of human chromosome 5 [87]. It was suggested that telomere looping formed by sufficiently long telomeres suppresses TERT expression by placing the shelterin factor TRF2 into the TERT promoter region [88]. In addition, the regulation of telomerase expression involves a number of epigenetic mechanisms, such as the modulation of the TERT promoter methylation, histone modification, and noncoding RNAs [89].

Data accumulated so far demonstrate that telomeres are under the strong influence of different endogenous and exogenous factors. Though the role of telomere shortening in replicative and chronological aging is beyond doubt [90], recent results highlight its role as a biosensor rather than a mitotic clock ensuring the accurate counting of cell doublings.

## 4. Replicative Senescence in Different Types of Stem Cells

Above all, aging is associated with the body’s inability to utilize nonfunctional senescent cells and replace them with new ones [91]. Both problems arise from the impairment of the proliferative activity and the inadequate differentiation capacity of stem cells, normally securing the physiological regeneration of tissues, including the renewal of the immune system. Since the differentiation potential of stem cells is tightly associated with their proliferation potential, progressive telomere shortening attenuates the regenerative capacity of stem cells [92]. In contrast, the recovery of the proliferative potential of stem cells restores their ability for multidirectional differentiations [93,94]. Therefore, stem cells link proliferative aging and the aging of the whole organism. The above-described ability of telomeres to react to different intracellular and external factors is especially important in stem cells. Stem cells can exist only in special tissue niches. The niche microenvironment can affect telomere length and, accordingly, the proliferative potential of stem cells. Renewal of the niche may induce rejuvenation of dwelling stem cells [95]. Telomerase expression is one more hallmark of stem cells. ESCs isolated from the blastocyst display high levels of telomerase activity [96]. Telomerase activity is present in most types of adult stem cells, though at substantially lower levels. Such lower levels are sufficient for slowing down telomere shortening and expanding the replicative lifespan [97] but cannot prevent replicative senescence. In this case, low telomerase expression may provide protection against the malignant transformation of stem cells [98].

Low oxygen content is typical for stem cell niches. Thus, in the bone marrow, niche hypoxia is the prerequisite of HSC maintenance in a quiescent state [99,100]. A low level of oxygen was reported in the NSC niche of the brain [101]. Hypoxia can limit the impact of oxidative stress on the telomeres and also stimulates their elongation and therefore enhances the proliferative potential of cells [102,103]. Hypoxia-inducible factor 1 (HIF1) is the main driver of this process, as well as of the majority of other processes related to the adaptation of cells to hypoxia. Under low-oxygen conditions, HIF1 activates the transcription of a number of genes responsible for the survival of cells and of the whole body in the low-oxygen environment. The HIF1-induced adaptation of cells to hypoxia involves the switching of the metabolism from oxidative phosphorylation to glycolysis, resulting in the inhibition of ROS production in the mitochondria [104,105]. HIF1 is able to bind to the TERT promoter region, directly inducing the activation of its expression [106]. The TERT promoter region carries specific binding sites for other stemness-related transcription factors, including STAT3 and MYC [107]. However, the telomerase expression level is associated with the proliferative activity of cells rather than with their stemness, and actively dividing cells can display higher TERT expression compared to stem cells [108]. In turn, TERT is involved in non-canonical functions unrelated to telomere elongation, such as the initiation of gene expression and chromatin remodeling [109]. For example, as a co-factor in the b-catenin complex, it is involved in the activation of the WNT pathway, which plays a major role in the regulation of stemness [110]. In the in vitro experiments, the hypoxia-induced enhancement of TERT expression contributes to keeping human ESCs in an undifferentiated state, without noticeable changes in telomerase activity [111]. The role of telomeres and telomerase in the maintenance of the homeostasis and functional activity of different types of stem cells is now considered further.

### 4.1. Embryonic Stem Cells

Pluripotent ESCs forming the blastocyst inner cell mass are the predecessors of all types of cells in the body. Not surprisingly, ESCs have long telomeres and exhibit high telomerase activity [96,112], making ESC cultures immortal. Moreover, the length of telomeres in animals cloned by SCNT is indistinguishable from the telomere length in normally born animals matched by age [113]. In iPSCs obtained by transfection with genes encoding for the Yamanaka factors (Oct4, Sox2, Klf4, c-Myc), telomeres are longer than in mother somatic cells [114,115,116]. Taken together, these data suggest the ability of pluripotent cells not only to maintain but also to extend the length of telomeres. Probably, at the blastocyst stage the pluripotent inner cell mass and epiblast cells restore the telomere length and, accordingly, recover their full proliferative potential. At the earlier cleavage stage of ontogenesis, the recombination-based alternative lengthening of telomeres (ALT) mechanism may be at work, as in some telomerase-negative tumors [117]. Later, at the blastocyst stage, telomerase activity is greatly enhanced, and, consequently, ALT elongation is not detected in human ESC cultures [118], though it occurs in mouse ESC lines [119] (Figure 4).

In mouse ESCs, the ALT takes place via telomere sister chromatid exchange-dependent homologous recombination and starts with the expression of the Zscan4 gene, expressed in the two-cell embryo and ESCs [120]. Zscan4 activation is responsible for the prolonged culture of telomerase-deficient late-generation murine ESCs and human ALT U2OS cancer cells with stably short telomeres [121]. Expression of this gene is enhanced upon telomere shortening and is controlled by several molecular actors, including the Rif1 telomere-associated protein, which also controls the level of other two-cell embryonic-specific factors. In mouse ESCs, Rif1 upholds the H3K9me3 histone methylation at the subtelomeric zones, thus inhibiting the expression of Zscan4 [122]. The expression of Zscan4 is positively controlled by Dcaf11 (Ddb1- and Cul4-associated factor 11) through the ubiquitination-mediated degradation of Kap1 (KRAB-associated protein 1), resulting in the switching on of the Zscan4 downstream enhancer and the removal of H3K9me3 histone at the subtelomeric regions [123]. Recently, it was confirmed that totipotent two-cell stage mouse blastomeres and murine ESCs maintain a robust transcriptional program, which includes high Zscan4 expression and ALT-like telomere extension [124].

Due to high telomerase activity, ESCs need to control the upper, rather than the lower, limit of telomere length, since exceedingly long telomeres can compromise genome stability. Telomere shortening is achieved by so-called telomere trimming [125]. The amount of available shelterin being constant, its concentration across the hyper-elongated telomere regions is reduced [126], allowing direct binding of the telomeric zinc finger-associated protein (TZAP) to TTAGGG repeats, resulting in the removal of the telomere part corresponding to the t-loop [127,128]. Telomere shortening initiates the extrachromosomal DNA t-circle release marking the process. Thus, the upper threshold of the telomere length, which will further determine the proliferative potential and differentiation capacity, is set up in early embryogenesis.

### 4.2. Germline Stem Cells

Since the germline cells constitute the material link between parents and progeny, maintenance of their stable telomere length across generations seems to be very important. However, the telomere length can be extended at the blastocyst, and probably also at the cleavage stage. Accordingly, even if telomeres are abnormally short in the zygote, they can regain regular length in the germline cells. In humans, primordial germ cells (PGCs) arise from the epiblast cells and initially are located within the yolk sac dorsal endoderm area [129]. Though PGCs do not produce somatic cell progeny, they express Oct4 and NANOG [130], suggesting that they are potentially pluripotent. Indeed, PGC-derived cell lines under specific culture conditions exhibit ESC-like differentiation properties [131]. In the course of gonadal development, PGCs migrate to the genital ridge to form the indifferent gonad and then undergo male or female sex specification [132].

Cells of the inner cell mass of the blastocyst can increase the length of their telomeres due to high telomerase activity. They probably control the upper limit of telomere length by trimming and can give rise to immortal cultures of embryonic stem cells (ESCs). PGCs arise from the epiblast cells and have properties similar to those of ESCs, including long telomeres and high telomerase activity. PGCs form a germline, which in adult males includes spermatogonial stem cells (SSCs). SSCs express telomerase and are characterized by either stable telomeres or weak telomere shortening that can be compensated later at the offspring blastocyst stage. Other types of adult stem cells, such as hematopoietic (HSC), mesenchymal (MSC), neural (NSC), and epithelial (EpSC), have shorter telomeres and generally low telomerase activity. They undergo replicative senescence, although more slowly than other actively proliferating somatic cells.

Male PGCs are localized in the testes, have long telomeres, high telomerase activity, and display multipotent stem cell differentiation potential [133]. After passing several intermediate differentiation stages, PGCs form a population of SSCs. This cell population exhibits the features of unipotent stem cells, since it is self-sustaining and gives rise to more differentiated spermatocytes and, finally, to spermatozoa. Unlike PGCs, SSCs do not express the pluripotency markers [130]. However, they continue to express telomerase [134]. Mouse SSC-derived cell lines retain the ability of spermatogenesis after transplantation into the testes. High proliferative potency allows them to proliferate for more than 2 years. However, their immortality is questionable since telomere shortening takes place during this period [135]. Telomere shortening is found in patients with oligospermia, a major cause of male infertility [136], while normally the telomere length in spermatozoa is substantially higher than in somatic cells.

Female PGCs located in the ovaries undergo step-by-step differentiation into the oogonia. After meiosis, oogonia become oocytes. Some authors claim the existence of oogonial stem cells in the gonads of adult women [137]. However, there is no hard evidence supporting this assumption [138]. The telomere length of immature oocytes exceeds that of mature oocytes [139]. Furthermore, telomerase activity can be detected during maturation but not in mature oocytes [140].

### 4.3. Hematopoietic Stem Cells

Telomerase activity is present in the multipotent HSCs, but its level is insufficient to completely prevent telomere erosion, resulting in the shortening of blood cell telomeres throughout a lifetime [141]. Telomerase activity has been found not only in HSCs but also in some of their more differentiated descendants. The ability of fast propagation of certain cell types in response to blood loss or infection is a characteristic functional feature of the hematopoietic system. To prevent crucial telomere shortening in actively proliferating cells, telomerase activation can occur in response to cytokine or antigen exposure [142]. This is most clearly seen in the example of the clonal expansion of lymphocytes. Upon contact with an antigen, a naïve lymphocyte acquires antigen specificity and further acts like a typical stem cell reproducing itself and forming a clone of more differentiated effector cells in order to sustain sensibility of the immune system to the antigen. Given this, it is no surprise that telomerase is active in both naïve and activated lymphocytes [143]. During aging, lymphocyte production is impaired because of the accumulation of SASP cells in the bone marrow stroma forming the HSC niche [144]. Because of that, aging causes a shift in HSC differentiation towards the myeloid lineage [145]. The impairment of the HSCs’ ability to support the telomere length leads to the development of pathological states, such as aplastic anemia, specific lymphopenias, or even total bone marrow failure [146,147].

### 4.4. Mesenchymal Stem Cells

Mesenchymal stem cells (MSCs) are multipotent cells with a fibroblast-like morphology that are capable of differentiation in at least three directions: osteogenic, chondrogenic, and adipogenic [148]. Originally, MSCs were isolated from bone marrow but were later found in other organs and tissues, including fat, dental pulp, endometrium, umbilical cord Wharton’s jelly, and others [149]. The efficacy of osteogenic differentiation of MSCs isolated from adults is diminished compared to MSCs isolated from children [150], suggesting that MSCs undergo aging in vivo. The proliferative potential of MSCs in culture is limited and close to the margin determined by Hayflick [151], and telomere shortening is due to cellular divisions rather than other age-related processes [152]. Telomerase activity in MSCs is low or altogether absent [108,153,154], and its varying level may be due to the heterogeneity of MSC populations [155]. In culture, MSCs gradually lose multipotency and homing ability but retain competence for adipogenic differentiation [156]. TERT overexpression makes MSCs immortal, sustains their differentiation and immunomodulation properties [157], and promotes tolerance to oxidative stress [158]. This provides favorable opportunities for telomerized MSC applications in regenerative medicine.

### 4.5. Neural Stem Cells

NSCs are oligopotent cells able to differentiate into neurons and glial cells, astrocytes, and oligodendrocytes. Like MSCs, transplanted NSCs exert therapeutic effects, especially in central nervous system disorders [159]. However, their applications are more restricted, since it is practically impossible to isolate them from adult donors, while the use of fetal NSCs is limited by ethical reasons and fears of teratomas.

Telomere shortening can interfere with the neuronal differentiation of NSCs [160]. TRF2, an essential component of the shelterin complex, slows down NSC differentiation by suppressing the transcriptional regulator REST in the self-renewing NSC subpopulation. The inhibition of REST induces down-regulation of a number of neuron-specific genes. NSC entry into differentiation is induced in the presence of the truncated isoform of TRF2 located in the cytoplasm and lacking the suppressive activity [161]. Remarkably, TRF2 disfunction and related genome instability, including chromosome fusion, has no impact on the functional activity of the affected terminally differentiated neurons in vivo [162]. Embryonic NSCs have high levels of telomerase expression [163]. The enzyme makes an important contribution to neurogenesis by enhancing the efficacy of neural differentiation [164]. The noncanonical telomerase function, unrelated to the support of the telomere length, also has an important impact, for instance, in spatial memory formation [165]. After the differentiation process has been initiated, the telomerase interferes even more with further differentiation at the progenitor cell stage, rather than helps it [166]. Notably, TERT has been detected in the cytoplasm of the mature neurons in the adult human hippocampus [167]. Probably, this kind of ectopic TERT expression protects neurons from apoptosis, as it happens in embryonic neurogenesis [168].

In the adult brain, NSCs are located in the neurogenic niches of the subgranular zone of the hippocampus dentate gyrus and the subventricular zone of the lateral ventricles. These niches exhibit telomerase activity but much less pronounced than in the embryonic NSCs [169]. In the rat brain, new neurons are produced constantly [170], and there is much evidence in favor of ongoing neurogenesis in the adult human brain [171,172]. However, recent studies have demonstrated that in the human hippocampus neurogenesis drops sharply over time, and in adulthood the production of new neurons is hard to detect [173]. The rate of the telomere attrition in the brain is slower than in other tissues, probably because of sluggish proliferation [174]. At the same time, many brain disorders, including cognitive disfunction, dementia, schizophrenia, and autism, are associated with critically short telomeres [175].

### 4.6. Tissue-Specific Epithelial Stem Cells

Epithelial cells are located at the border of tissues and the environment and are constantly exposed to negative external influences. Therefore, the presence of epithelial stem cells (EpSCs) possessing proliferative potential, ensuring rapid renewal and repair of the epithelium, is a prerequisite of its normal functioning. Usually, EpSCs have longer telomeres compared to other cells of the same tissue, and this can be applied to the search for tissue stem cell niches [176].

Normal aging is associated with skin atrophy and the loss of hair follicles because of epidermal stem cell dysfunction and telomere shortening [177]. Stem cells causing the renewal of skin epidermis persist in the stratum basale during a lifetime. They slowly proliferate and produce more actively proliferating stem cells, which in turn give rise to rapidly dividing progenitor cells, finally differentiating into keratinocytes [178]. EpSCs express the stem cell hallmarks, including genes regulating the telomerase activity [179]. Interestingly, TERT can directly activate quiescent hair follicle stem cells via its noncanonical activity in the absence of TERC [180]. Nevertheless, patients with dyskeratosis congenita, who have shortened telomeres because of a TERC mutation, are subject to skin atrophy and premature hair graying [181]. Their keratinocytes are characterized by a reduced potential for proliferation and colony formation in vitro [182]. In contrast, in patients with Werner’s syndrome (mutation of the WRN helicase gene) displaying analogous symptoms, keratinocytes have normal proliferative potential, while the in vitro proliferation potential of their skin fibroblasts is reduced [183]. The poor functioning of EpSCs in patients with progeroid disorders, unrelated directly to the mutations in genes encoding for the telomerase complex factors, may be associated with the failure of mesenchymal cells to adequately support epidermal cell growth [184]. A closer insight into the molecular mechanisms of the dermal epithelium stem cell suppression caused by the telomere dysfunction revealed the upregulation of the Follistatin protein, a Smad signaling inhibitor which blocks skin stem cells, including hair follicle stem cell differentiation, via the BMP regulatory pathway [185].

Quiescent and actively proliferating stem cells were identified in the intestine [186,187]. They are located in the basal part of crypts and, as differentiation progresses, migrate towards the villi, where they give rise to enteroendocrine cells, tuft cells, goblet cells, M cells, and Paneth cells [188]. Thus, unlike skin unipotent EpSCs, intestinal stem cells are oligopotent and can differentiate into different cell types present within the tissue. Intestinal epithelium is the fastest self-renewing tissue with a cell turnover time of about 4–5 days [189]. Not surprisingly, intestinal crypts, which home the stem cell niches, display telomerase activity [190]. It was reported that active telomerase is confined to the slowly proliferating stem cells, which undergo asymmetric divisions reproducing themselves and yielding Lrg-positive stem cells lacking telomerase activity [186]. The Lrg-positive stem cells divide symmetrically and, while moving apically towards the villi, generate a progenitor cell population. These data contest the opinion that more actively proliferating cells, including the Lrg-positive intestinal stem cells, have higher levels of telomerase expression. However, during the mouse lifespan an intestinal stem cell and its progeny must go through 700–1000 divisions, suggesting crucial telomere shortening in the absence of telomerase activity in actively dividing cells [191]. It should be taken into account that in intestinal epithelial cells the telomere attrition is dependent upon intratissue and external factors specific to the intestine. In this organ, stem cells are in complex interactions with the mesenchymal and immune cells controlling local inflammatory reactions [192] as well as with the microbiota acting through ROS production or otherwise [193].

Aging of the regional stem cells and associated telomere shortening are involved in the pathogenesis of a number of lung diseases, including idiopathic pulmonary fibrosis, chronic obstructive pulmonary disease, SARS-CoV-2 infection, and lung cancer [194]. Lung tissue contains several types of poorly characterized EpSCs, located mainly in the distal parts of the bronchi [195,196]. Unlike them, type 2 alveolar epithelial cells (AEC2s) reside within the alveolar epithelium and have been studied much better because they are the prime target of the SARS-CoV-2 virus. AEC2s constitute 15% of the alveolar epithelium and are capable of self-renewal. Type 1 alveolar epithelial cells (AEC1s), the cell type predominant in the alveolar epithelium, are the product of proliferation and the consecutive terminal differentiation of AEC2s. It has been suggested that AEC2 aging leads not only to impaired AEC1 replenishment but also to aberrant transdifferentiation and pulmonary fibrosis [197,198]. Interestingly, in addition to AEC2 depletion, the SARS-CoV-2 virus can impair alveolar regeneration by the induction of AEC2 proliferative aging [199]. The molecular mechanism underlying AEC2 aging induction involves the TGF-β signaling pathway inducing suppression of the TERT expression, as well as the shelterin TPP1 degradation-mediated DNA damage response. TPP1 degradation can be associated with different environmental stress factors, including smoking, bacterial toxins, and ionizing radiation [197].

The possible role of the regional EpSCs in the initiation of solid tumors is one of the significant problems related to their aging [200]. These long-living cells have the proliferative potential and activity required to accumulate oncogenic mutation in numbers sufficient for malignant transformation. Proliferative senescence and telomere erosion can lead to genomic instability, substantially enhancing mutagenesis. The ensuing cancer stem cells can undergo asymmetrical divisions, supporting a pool of tumor cells highly resistant to therapy and capable of initiating de novo tumor growth.

## 5. Conclusions

It is evident now that telomeres cannot serve as an accurate mitotic clock, since their length and, accordingly, their cell aging are under the influence of various internal and external factors. The telomere-centric theory of aging has in fact integrated the alternate stochastic theories, including the ROS theory claiming the central role of the free radicals [201]. Stem cells participating in the development of a new organism undergo rebooting of their epigenetic profile and telomere length [3], thus ensuring genome stability across generations (Figure 4). On the other hand, adult stem cells responsible for the physiological and reparative regeneration of tissues undergo proliferative senescence, despite the molecular mechanisms limiting its impact. A low proliferative rate and asymmetric mitoses allow the long-term maintenance of quiescent stem cells, while their actively dividing progeny exhaust the proliferative potential. In addition, adult stem cells are highly resistant to external factors inducing telomere erosion and display certain telomerase activity, albeit insufficient for the full recovery of the telomeres. Compared to earlier ideas about the mitotic clock running until the alarm turns on [202], the contemporary views provide far more opportunities to develop novel approaches to slowing down stem cell aging and, consequently, the aging of the whole organism. Those approaches include but are not limited to pharmacological drugs, such as antioxidants, changing nutrition and behavior, and avoiding environmental pollutants that stimulate telomere attrition.

The question of whether the Hayflick limit can impede the manufacture of cell-based products for biomedicine is still under discussion. Simple calculations show that the total cell mass that can be obtained in culture under conditions similar to those in the Hayflick experiments (55 doublings with an initial cell number of 10^6^) [41] is several tens of millions of metric tons. Such a yield would satisfy any manufacturer. However, these calculations have little to do with reality. The use of fetal cells, as Hayflick did in his experiments, is limited for ethical reasons and completely impossible for autologous cell manufacturing. Commonly used cell cultures have significantly lower proliferative potential. In addition, stem cell cultures may be heterogeneous when some cells differentiate or undergo apoptosis, thereby reducing the overall yield. In this regard, the lack of proliferative potential can become a real problem in the large-scale production of cells. However, approaches aimed at reducing telomere erosion and stimulating endogenous telomerase activity can significantly increase the replicative potential of cultured stem cells. Maintaining cells in low-oxygen conditions [203,204] or in the presence of hydrogen gas [205], matrix modification [206], and supplying the culture medium with growth factors [207] and antioxidants capable of attenuating ROS accumulation [208] can slow done the telomere shortening and proliferative senescence.

A better way to achieve the required yield in the production of cultured cells is the immortalization of cells by introducing the TERT gene. This approach continues to raise concerns about the possibility of malignant transformation, but there is evidence that they may be exaggerated. Expression of exogenous TERT in cells not only fails to cause malignant transformation [209] but also apparently increases the genetic stability of aging cells and improves the control over their proliferation [210]. Importantly, overexpression of telomerase does not alter stem cell differentiation [211]. The development of novel genetic engineering technologies, including genomic editing using the CRISPR/Cas9, will likely enable the production of safe vector systems, for example, vectors capable of preprogrammed self-destruction [212]. This will make it possible to express transgenic telomerase at the stage of cell expansion in vitro and to switch it off before cell transplantation, significantly reducing doubts about the safety of using telomerase in cell-based therapy protocols.

## Figures and Tables

**Figure 1 biomedicines-10-02335-f001:**
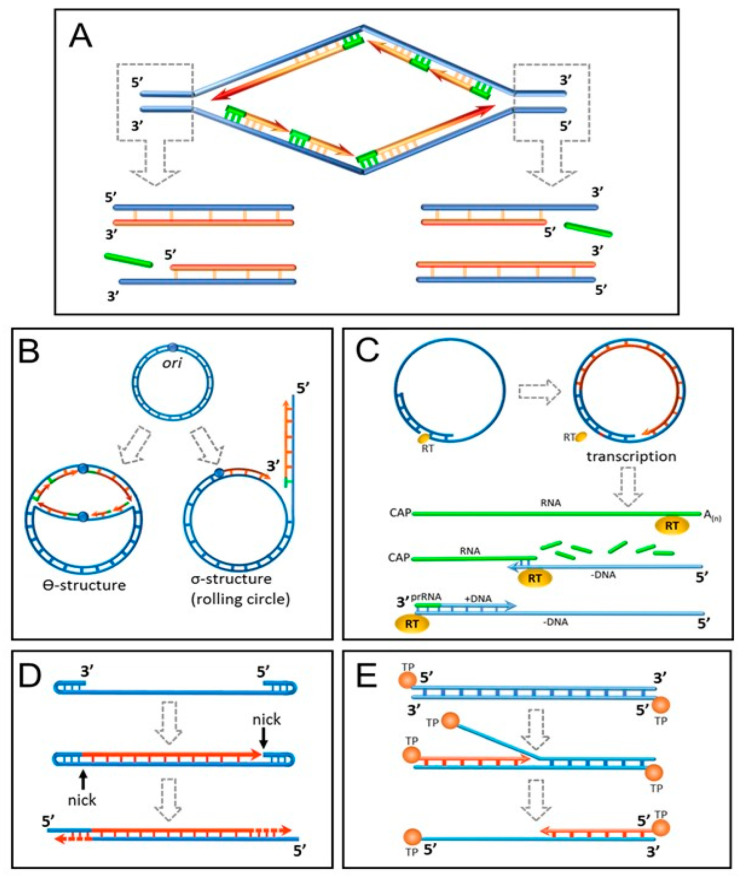
The problem of incomplete DNA replication and its solution by “primitive” organisms. (**A**) Incomplete replication of the 3′ end of DNA in eukaryotic cells. Replicons located in the telomere region cannot displace the last RNA primer of the lagging strand. (**B**) A covalently closed circular genome replication. In theta replication (prokaryotes, viruses), the last RNA primer of the lagging strand is replaced by the leading strand. A rolling cycle mechanism (viruses, plasmids) uses a single-stranded nick as a primer for DNA synthesis. As a result, a concatemer containing several complete genome copies is formed. (**C**) Hepadnaviruses do not have a covalently closed circular genome, but it can be easily restored by reparative DNA synthesis after cell infection. Despite this, the virus uses reverse transcription as a strategy for its genome replication. Whole genome RNA, after its translation, is used as a template for minus DNA strand synthesis. Reverse transcriptase (RT) includes a domain that can prime DNA synthesis from a tyrosine residue. The incomplete plus DNA strand is synthesized by DNA polymerase primed with the rest of the degraded whole genome RNA. (**D**) Parvoviruses use hairpins at the ends of their DNA (rabbit ears) as primers for DNA polymerase. (**E**) Adenovirus DNA polymerase forms a complex with a terminal protein (TP) that can act as a primer for unidirectional DNA replication.

**Figure 2 biomedicines-10-02335-f002:**
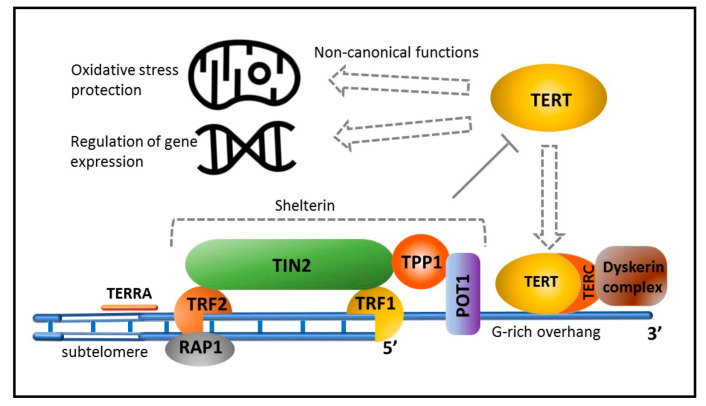
The main players of telomere stabilization and elongation. Shelterin contains six proteins that stabilize chromosome ends, prevent activation of the DNA damage response (DDR) system, and suppress telomerase-dependent telomere elongation. In the absence of fully assembled shelterin, telomerase-mediated lengthening of telomeres becomes possible and proceeds via the binding of TERT to TERC stabilized by the dyskerin complex (dyskerin, NOP10, NHP2, GAR1). TERT can also exhibit non-canonical activity by modulating gene expression to protect against cell death following double-stranded DNA damage and protect mitochondria from ROS.

**Figure 3 biomedicines-10-02335-f003:**
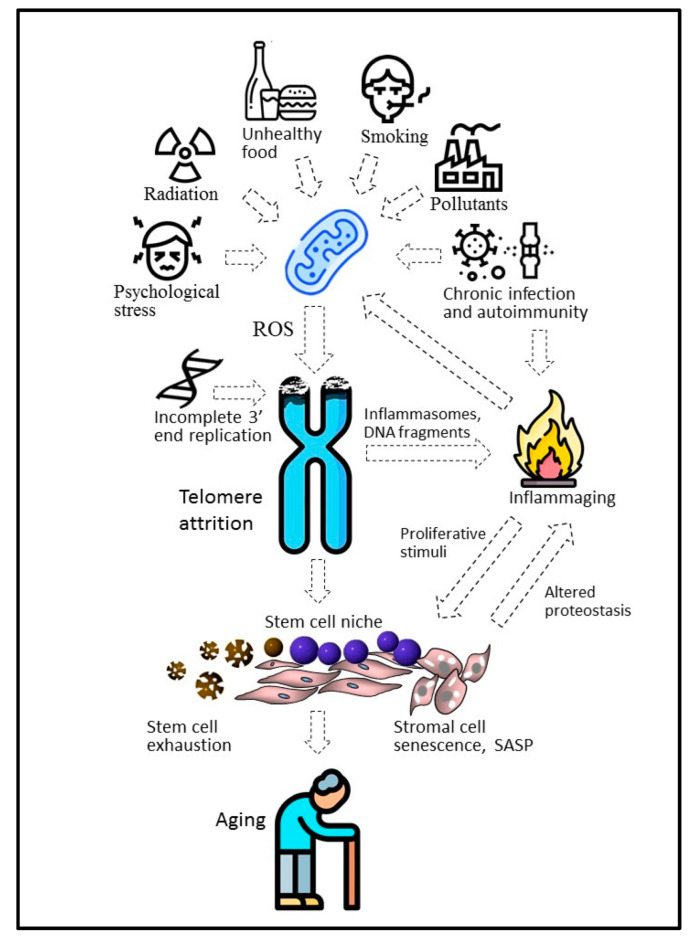
Telomere theory of aging; state of the art.

**Figure 4 biomedicines-10-02335-f004:**
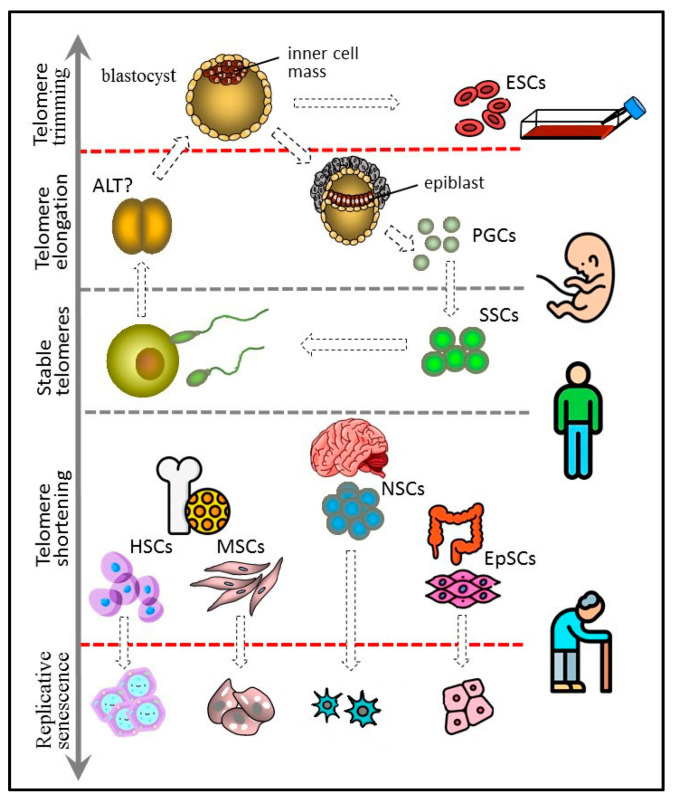
Telomere length dynamics in various types of stem cells.

## Data Availability

Not applicable.

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
