# Peer review of "Telomeres and Telomerase in the Control of Stem Cells"

_biomedicines, 2022, doi:10.3390/biomedicines10102335_

Round 1
Reviewer 1 Report
The text is clearly written and well organized. It will be very useful for students for all the mains finding in the field summarized. The only objection the reviewer has - the legend under Figure 1 is of the same fond and lines' space as the main text. Makes it difficult to comprehend. Please, correct.
Author Response
Dear Reviewer,
Thank you very much for the analysis of our manuscript and pointing out the flaw. We amended the legend of Figure 1 accordingly.
Reviewer 2 Report
The article by Lupatov AY and Yarygin KN is a detailed review on knowledge on telomeres and telomerase in non pathological stem cells. This review was sub-divided in different paragraphs first describing the different types of stem cells (adult or embryonic) and then the mechanisms and actors involved in regulation of telomere attrition or extention with an historical, very pertinent analysis. The importance of microenvironment parameters like oxygen concentration and stress/ aging are also well described and then an highlight on knowledge on telomerase activity and regulation of telomere length in these different types of stem cells (embryonic, germline, hematopoietic, mesenchymal, neural and epithelial) was given.
This review is well written and of interest but some improvements should be done.
1) The paragraph N°2, “what are stem cells” should be shorten and better organized with the more precise reminder of the knowledge of differences between adult versus embryonic/ iPSC for telomerase activity and regulation of telomere length.
2) A scheme with the main actors of telomere elongation and of the process of telomerase activity to elongate telomeres should be included and appreciated.
3) The recent publications on ALT (Alternative Lengthening of Telomeres) mechanism, mainly described in embryonic stem cells, should be cited and commented since the identification of new players in the process, independent of telomerase activity, open new roads to better understand the regulation of chromosome stability and telomere functions and elongation/ attrition.
Articles to cite and to comment:
Zalzman M et al (2010): Zscan4 regulates telomere elongation and genomic stability in ES cells. Nature. Apr 8;464(7290):858-63. doi: 10.1038/nature08882. Epub 2010 Mar 24. PMID: 20336070.
Le R. et al, (2021): Dcaf11 activates Zscan4-mediated alternative telomere lengthening in early embryos and embryonic stem cells. Cell Stem Cell, Apr 1;28(4):732-747.e9.
Markiewicz-Potoczny M et al. (2021): TRF2-mediated telomere protection is dispensable in pluripotent stem cells. Nature. 2021 Jan;589(7840):110-115. doi: 10.1038/s41586-020-2959-4. Epub 2020 Nov 25. PMID: 33239785.
Dan J et al. (2022): Zscan4 Contributes to Telomere Maintenance in Telomerase-Deficient Late Generation Mouse ESCs and Human ALT Cancer Cells. Cells. 2022 Jan 28;11(3):456. doi: 10.3390/cells11030456. PMID: 35159266
Minor concerns:
References N°15, 124 are impossible or difficult to find and should be deleted.
Ref. 141, 151, 154, 172, 177, 203 should be double checked.
At line 375, the comma should be removed: “At the earlier (no comma) cleavage….”
